# SELF-SUPERVISED CONTINUAL LEARNING BASED ON BATCH-MODE NOVELTY DETECTION

## ABSTRACT

Continual learning (CL) plays a key role in dynamic systems in order to adapt to new tasks, while preserving previous knowledge. Most existing CL approaches focus on learning new knowledge in a supervised manner, while leaving the data gathering phase to the novelty detection (ND) algorithm. Such presumption limits the practical usage where new data needs to be quickly learned without being labeled. In this paper, we propose a unified approach of CL and ND, in which each new class of the out-of-distribution (ODD) data is first detected and then added to previous knowledge. Our method has three unique features: (1) a unified framework seamlessly tackling both ND and CL problems; (2) a self-supervised method for model adaptation, without the requirement of new data annotation; (3) batch-mode data feeding that maximizes the separation of new knowledge vs. previous learning, which in turn enables high accuracy in continual learning. By learning one class at each step, the new method achieves robust continual learning and consistently outperforms state-of-the-art CL methods in the single-head evaluation on MNIST, CIFAR-10, CIFAR-100 and TinyImageNet datasets.

## 1 INTRODUCTION

Machine learning methods have been widely deployed in dynamic applications, such as drones, self-driving vehicles, surveillance, etc. Their success is built upon carefully handcrafted deep neural networks (DNNs), big data collection and expensive model training. However, due to the unforeseeable circumstances of the environment, these systems will inevitably encounter the input samples that fall out of the distribution (OOD) of their original training data, leading to their instability and performance degradation. Such a scenario has inspired two research branches: (1) novelty detection (ND) or one-class classification and (2) continual learning (CL) or life-long learning. The former one aims to make the system be able to detect the arrival of OOD data. The latter one studies how to continually learn the new data distribution while preventing catastrophic forgetting Goodfellow et al. (2013) from happening to prior knowledge.

While there exists a strong connection between these two branches, current practices to solve them are heading toward quite different directions. ND methods usually output all detected OOD samples as a **single class** while CL methods package **multiple classes** into a single task for learning. Such a dramatic difference in problem setup prevents researchers to form an unified algorithm from ND to CL, which is necessary for dynamic systems in reality.

One particular challenge of multi-class learning in CL is the difficulty in data-driven (i.e., self-supervised or unsupervised) separation of new and old classes, due to their overlap in the feature space. Without labeled data, the model can be struggling to find a global optimum that can successfully separate the distribution of all classes. Consequently, the model either ends up with the notorious issue of catastrophic forgetting, or fails to learn the new data. To overcome this challenge, previous methods either introduce constraints in model adaptation in order to protect prior knowledge when learning a new task Aljundi et al. (2018); Li & Hoiem (2018); Kirkpatrick et al. (2017); Rebuffi et al. (2017), or expand the network structure to increase the model capacity for new knowledge Rusu et al. (2016); Yoon et al. (2017). However, the methods with constraints may not succeed when the knowledge distribution of a new task is far from prior knowledge distribution. On the other hand, the methods with a dynamic network may introduce too much overhead when the amount of new knowledge keeps increasing.

In this context, our aims of this work are: (1) Connecting ND and CL into one method for dynamic applications, (2) Completing ND and CL without the annotation of OOD data, and (3) Improving the

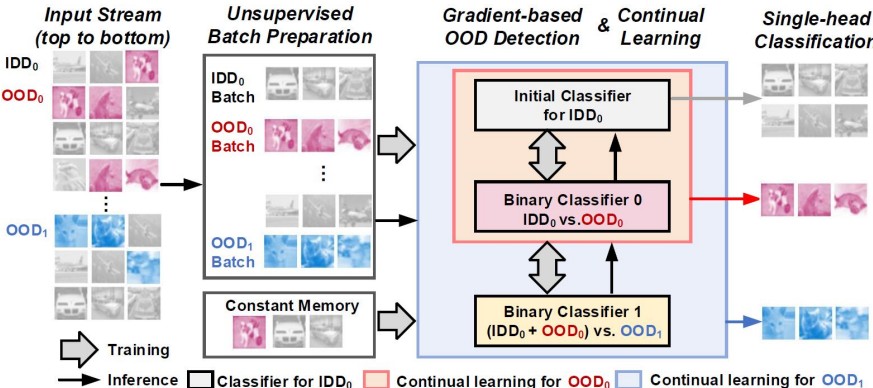

Figure 1: The framework of our unified method for both novelty detection and one-class continual learning. The entire process is self-supervised, without the need of labels for new data.

robustness and accuracy of CL in the single-head evaluation. We propose a self-supervised approach for one-class novelty detection and continual learning, with the following contributions:

- A unified framework that continually detects the arrival of the OODs, extracts and learns the OOD features, and merges the OOD into the knowledge base of previous IDDs. More specially, we train a tiny binary classifier for each new OOD class as the feature extractor. The binary classifier and the pre-trained IDD model is sequentially connected to form a "$N + 1$" classifier, where "$N$" represents prior knowledge which contains $N$ classes and "1" refers to the newly arrival OOD. This CL process continues as "$N + 1 + 1 + 1...$" (i.e, one-class CL), as demonstrated in this work.

- A batch-mode training and inference method that fully utilizes the context of the input and maximizes the feature separation between OOD and previous IDDs, without using data labels. This method helps achieve the high accuracy in OOD detection and prediction in a scenario where IDDs and OODs are streaming into the system, such as videos and audios.

- Comprehensive evaluation on multiple benchmarks, such as MNIST, CIFAR-10, CIFAR-100, and TinyImageNet. Our proposed method consistently achieves robust and a high single-head accuracy after learning a sequence of new OOD classes one by one.

## 2 BACKGROUND

Most continual learning methods belong to the supervised type, where the input tasks are well-labeled. To mitigate catastrophic forgetting, three directions have been studied in the community: (1) **Regularization** methods Zeng et al. (2019); Aljundi et al. (2018); Li & Hoiem (2018); Kirkpatrick et al. (2017); Rebuffi et al. (2017); Zenke et al. (2017); Ahn et al. (2019), which aim to penalize the weight shifting towards the new task. This is realized by introducing a new loss constraint to protect the most important weights for previous tasks. Many metrics to measure the importance of weights are proposed, such as the Fisher information matrix, distillation loss and training trajectory. (2) **Rehearsal-based** methods Lopez-Paz & Ranzato (2017); Chaudhry et al. (2018); Rolnick et al. (2019); Aljundi et al. (2019); Cha et al. (2021), which maintain a small buffer to store the samples from previous tasks. To prevent the drift of prior knowledge, these samples are replayed during the middle of the training routine on the new task. Some rehearsal-based methods are combined with the regularization methods to improve their performance. (3) **Expansion-based** methods Rusu et al. (2016); Yoon et al. (2017); Schwarz et al. (2018); Li et al. (2019); Hung et al. (2019), which aim to protect previous knowledge by progressively adding new network branches for the new task. These methods are especially useful when the knowledge base of previous tasks and that of the new task are overlapping. Although these three supervised methods improve the performance of continual learning, we argue that the capability to automatically detect the task shifting and learn the new knowledge (i.e., self-supervised or unsupervised) is the most preferred solution by a realistic system, since an expensive annotation process of the new task is impractical in the field.

There also exist self-supervised CL methods Rao et al. (2019); Smith et al. (2021), which focus on the generation of the class-discriminative representations from the unlabelled sequence of tasks. However, their demonstration so far are still limited to relatively simple and low-resolution datasets, such as MNIST and CIFAR-10.

## 3 METHODOLOGY

### 3.1 TERMINOLOGY

**Algorithm 1** Unification of One-class Novelty Detection and Continual Learning

---

**Input:** IDD memory budget $\mathcal{X}_{IDD}^{mem} = \{X_{IDD}^{mem_1}, \ldots, X_{IDD}^{mem_N}\}$, IDD sequence $\mathcal{X}_{IDD} = \{X_{IDD}^1, \ldots, X_{IDD}^N\}$ OOD sequence $\mathcal{X}_{OOD} = \{X_{OOD}^0, \ldots, X_{OOD}^t\}$, trained model $\mathcal{M}_0$

1: $\mathcal{M}_{current} \leftarrow \mathcal{M}_0$
2: **for** $i = 0$ to $len(\mathcal{X}_{OOD})$ **do**
3: $\quad X_{OOD}^{current} \leftarrow \mathcal{X}_{OOD}[i]$
4: $\quad \mathcal{X}_{mixture} \leftarrow \mathcal{X}_{IDD} + X_{OOD}^{current}$
5: $\quad \mathcal{D}_{IDD} \leftarrow IDD\_estimator(\mathcal{X}_{IDD}^{mem}, \mathcal{M}_{current})$
6: $\quad X_{OOD}^{pred}, \mathcal{B}_i \leftarrow noveltyDetector(\mathcal{M}_{current}, \mathcal{X}_{IDD}^{mem}, \mathcal{D}_{IDD}, X_{mixture})$
7: $\quad \mathcal{X}_{IDD}^{mem}.append(X_{OOD}^{pred})$
8: $\quad \mathcal{M}_{current} \leftarrow \mathcal{M}_{current} + \mathcal{B}_i \quad //Merge\ Model$
9: **end for**
10: **return** $\mathcal{M}_{current}$

---

Previous CL methods use a task-based setup, where each task consists of multiple classes of training samples. The model is trained to learn each task (i.e., multiple classes) sequentially. Different from previous methods, our proposed solution is an unified system that leverages the output from the novelty detector for CL. Therefore, we embrace the one-class per task setup as follows:

As shown in Algorithm 1, the system is continuously exposed to a stream of mixed input that contains both in-distribution data (IDD) $\mathcal{X}_{IDD} = \{X_{IDD}^1, \ldots, X_{IDD}^N\}$ and one of the out-of-distribution (OOD) data $X_{OOD}^{current} \in \mathcal{X}_{OOD} = \{X_{OOD}^0, \ldots, X_{OOD}^t\}$, where each $X^i$ corresponds to a single class in the dataset. $\mathcal{X}_{IDD}$ denotes the IDD set containing all the classes that the system already recognizes. $\mathcal{X}_{OOD}$ contains the OOD classes that the system is currently facing and will encounter in the future. The primary task is to filter out $X_{OOD}^{current}$ from $\mathcal{X}_{IDD}$ through the novelty detection engine, while learning the features of $X_{OOD}^{current}$ for continual learning. Once $X_{OOD}^{current}$ is successfully detected and learned, we move $X_{OOD}^{current}$ from $\mathcal{X}_{OOD}$ into $\mathcal{X}_{IDD}$, and randomly draw a new OOD class from $\mathcal{X}_{OOD}$ as the new $X_{OOD}^{current}$. This process continues until $\mathcal{X}_{OOD}$ is empty and all classes have been learned by the model, i.e., becoming IDDs. For simplicity, we denote the initial IDD classes and the first OOD class as $\mathcal{X}_{IDD}^0$ and $X_{OOD}^0$, respectively. When the system has successfully learned $X_{OOD}^0$, we denote the updated IDD set as $\mathcal{X}_{IDD}^1$ and the next OOD as $X_{OOD}^1$. Then $\mathcal{S}_i = \{\mathcal{X}_{IDD}^i, X_{OOD}^i\}$ represents the status when the system just finishes learning $X_{OOD}^{i-1}$. From the perspective of task-based learning, each $X_{OOD}^i$ can be considered as a single task. Our proposed method will detect and learn only one class at each time.

### 3.2 ONE-CLASS NOVELTY DETECTION AND CONTINUAL LEARNING

Our method is built upon the recently proposed work on "Self-supervised gradient-based novelty detector" Sun et al. (2022). The contributions of this method are two-fold. First, given a pre-trained model $\mathcal{M}_0$ that is able to classify $\mathcal{X}_{IDD}^0$, a statistical analysis evaluating the Mahalanobis distance in the gradient space is developed to threshold the OOD. Second, to further boost the performance, they introduce a self-supervised binary classifier, denoted as $\mathcal{B}_0$, which guides the label selection process to generate the gradients so that the Mahalanobis distance between the IDD and OOD data is further maximized. The primary OOD detector, which is based on the Mahalanobis distance, interleaves with the binary classifier. As more data stream into the system, the OOD detection accuracy gradually improves through this closed-loop interaction.

This method can be naturally converted into a one-class CL solution. Upon the successful detection of $X_{OOD}^0$, the binary classifier $\mathcal{B}_0$ is well-trained to recognize $X_{OOD}^0$ from previous $\mathcal{X}_{IDD}^0$. Intuitively, we can deploy $\mathcal{B}_0$ upstream from the pre-trained IDD model $\mathcal{M}_0$ to filter out the newly learned $X_{OOD}^0$, and leave the predicted $\mathcal{X}_{IDD}^0$ samples to the downstream $\mathcal{M}_0$ for further classification. This merged model $\mathcal{M}_1 = \{\mathcal{M}_0 + \mathcal{B}_0\}$ will become the new baseline to detect the next $X_{OOD}^1$. Continuously, we can keep adding a new binary classifier $\mathcal{B}_i$ to the previously merged model $\mathcal{M}_i = \{\mathcal{M}_{i-1} + \mathcal{B}_{i-1}\}$ for every new $X_{OOD}^i$ upon its detection. Eventually, we obtain this

chain structure (Fig. 1) where the latest binary classifier is the most knowledgeable one to recognize all $\mathcal{X}_{IDD}$ and $\mathcal{X}_{OOD}$. During the inference phase, its job is to filter out the $X_{OOD}^t$ from a mixture input that contains $\{\mathcal{X}_{IDD}^t, X_{OOD}^t\}$. The remaining inputs $\mathcal{X}_{IDD}^t = \{\mathcal{X}_{IDD}^{t-1}, X_{OOD}^{t-1}\}$ will be sent downstream to the next binary classifier to sequentially recognize the $X_{OOD}^{t-1}$. This sequence continues until only $\mathcal{X}_{IDD}^0$ is left and arrives at the original model $\mathcal{M}_0$ for final classification.

---

**Algorithm 2** Gradient-based Novelty Detection

---

1: **function** $noveltyDetector\ (\mathcal{M}, \mathcal{X}_{IDD}^{mem}, \mathcal{D}_{IDD}, X_{mix})$
2:     Initialize $\mathcal{B}_{new}$
3:     $X_{mix}^{pure} \leftarrow batchPurityEstimator(X_{mix}, \mathcal{X}_{IDD}^{mem}, \mathcal{M})$
4:     **repeat** 10 iterations
5:         $X_{IDD}^{pred} \leftarrow [\ ], X_{OOD}^{pred} \leftarrow [\ ]$
6:         **for each** $x\ \in\ X_{mix}^{pure}$ **do**
7:             **if** $\mathcal{B}_{new}\ not\ trained$ **then**
8:                 $c_{\mathcal{M}_0} \leftarrow c_{\mathcal{M}_0}^{pred}, c_{\mathcal{B}_i} \leftarrow c_{\mathcal{B}_i}^{pred}$
9:             **else**
10:                $c_{\mathcal{M}_0} \leftarrow c_{\mathcal{M}_0}^{cust}, c_{\mathcal{B}_i} \leftarrow 1 - c_{\mathcal{B}_i}^{pred}$
11:            **end if**
           $\nabla f_{\mathcal{M}}(x) \leftarrow \nabla_{c_{\mathcal{M}_0}} f_{\mathcal{M}_0}(x) \| \dots \| \nabla_{c_{\mathcal{B}_i}} f_{\mathcal{B}_i}(x)$
12:            $noveltyScore\_l \leftarrow [\ ]$
13:            **for each** $(\hat{\mu}_c, \hat{\Sigma}_c)$ in $\mathcal{D}_{IDD}$ **do**
           $score\_c \leftarrow (\nabla f_{\mathcal{M}}(x) - \hat{\mu}_c)^T \hat{\Sigma}_c^{-1} (\nabla f_{\mathcal{M}}(x) - \hat{\mu}_c)$
           $noveltyScore\_l.append(score\_c)$
14:            **end for**
15:            $noveltyScore \leftarrow \mathbf{min}(noveltyScore\_l)$
16:            **if** $noveltyScore < threshold$ **then**
17:                $X_{IDD}^{pred}.append(x)$
18:            **else**
19:                $X_{OOD}^{pred}.append(x)$
20:            **end if**
21:        **end for**
22:        Re-initial & Train $\mathcal{B}_{new}$ using $\{X_{IDD}^{pred}, X_{OOD}^{pred}\}$
23:    **return** $\mathcal{X}_{IDD}^{pred}, \mathcal{B}_{new}$
24: **end function**

---

There are two advantages of this one-class learning method. First, after we merge a new binary classifier to the existing structure, all weights are frozen and isolated from future training towards the new OODs. This prevents the knowledge shifting and thus, minimizes catastrophic forgetting. Second, each binary classifier only induces a small memory overhead since its task is simple enough to be accomplished with very few layers of neurons. Compared with previous dynamic methods which either add an indeterminate amount of neurons to each layer or make new branches using sub-modules, our method requires much lower memory.

On the other hand, such a sequential learning model leaves two questions to be answered: (1) How to port the original Mahalanobis distance method into this merged model so that it can detect the next OOD? (2) How to achieve a high inference accuracy of the binary classifier, such that the testing samples can go through multiple steps in this sequence and reach the final classifier without losing the accuracy. We will address each problem in the subsection 3.3 and 3.4, respectively.

### 3.3 GRADIENT-BASED NOVELTY DETECTOR

The stepping stone of the previous novelty detector (Algorithm 2, Algorithm 3) is to characterize the IDD and OOD in the gradient space. If a model $\mathcal{M}$ has been trained to classify $N$ classes in $\mathcal{X}_{IDD}^{train} = \{X_{IDD}^{train-1}, \dots, X_{IDD}^{train-N}\}$, the gradients collected from each $X_{IDD}^i \in \mathcal{X}_{IDD}$ will form a class-wise multi-variant distribution $\mathcal{D}_{IDD} = \{D_{IDD}^1, \dots, D_{IDD}^N\}$. Each $D_{IDD}^i \sim \mathcal{N}(\hat{\mu}_i, \hat{\Sigma}_i)$ corresponds to a Gaussian distribution for a particular class $i$, and future IDDs $\mathcal{X}_{IDD}^{val} = \{X_{IDD}^{val-1}, \dots, X_{IDD}^{val-N}\}$ will be within the range of corresponding $D_{IDD}^i$. On the contrary, due to the low confidence towards the OOD, the model $\mathcal{M}$ will adjust itself to fit the OOD properly by back-propagating the gradients with abnormal magnitude and direction. Any deviant from $\mathcal{D}_{IDD}$ will be considered as abnormal and therefore, a distance metric can be utilized to measure the novelty confidence. They propose to use the Mahalanobis Distance as the novelty confidence score Sun et al. (2022):

$$M_x = (\nabla_c f_{\mathcal{M}}(x) - \hat{\mu}_c)^T \hat{\Sigma}_c^{-1} (\nabla_c f_{\mathcal{M}}(x) - \hat{\mu}_c) \tag{1}$$

To collect the gradient $\nabla_c f_{\mathcal{M}}(x)$, they further utilize two types of labels for back-propagation: $c_{\mathcal{M}}^{predicted}$ and $c_{\mathcal{M}}^{custom}$.

$$c_{\mathcal{M}}^{predicted} = \underset{c \in \mathcal{X}_{IDD}}{argmax}(Softmax(f_{\mathcal{M}}(X; \Theta))) \tag{2}$$

$$c_{\mathcal{M}}^{custom} = \underset{c \in \mathcal{X}_{IDD}}{argmin}(Softmax(f_{\mathcal{M}}(X;\Theta))) \tag{3}$$

Measured by the softmax output from the trained model $\mathcal{M}$, $c_{\mathcal{M}}^{predicted}$ and $c_{\mathcal{M}}^{custom}$ refer to the most and least possible class the input belongs to, respectively, leading to mild and aggressive back-propagated gradients. Since the goal is to detect the abnormal gradient from the OOD, using $c_{\mathcal{M}}^{custom}$ for OOD maximizes weight update to the model and thus, the gradients will be even more likely to fall out of the $\mathcal{D}_{IDD}$. On the other hand, using $c_{\mathcal{M}}^{predicted}$ for IDDs helps their gradients stay within $\mathcal{D}_{IDD}$ and thus, reduces the possibility of false alarm. To guide the label selection, they introduce a binary classifier into the system to predict the samples. Based on the prediction, the primary ND engine $\mathcal{M}$ collects the gradients with either $c_{\mathcal{M}}^{predicted}$ or $c_{\mathcal{M}}^{custom}$ to measure the Mahalanobis distance towards $\mathcal{D}_{IDD}$.

To make the above algorithm compatible with the unified ND model, we set up the following scenario for explanation. Assume the system just finishes the detection and learning on $X_{OOD}^i$ and is in the status of $\mathcal{S}_{i+1} = \{\mathcal{X}_{IDD}^{i+1}, X_{OOD}^{i+1}\}$. The latest binary classifier $\mathcal{B}_i$, which can distinguish $X_{OOD}^i$ and $\mathcal{X}_{IDD}^i$, is placed upstream from the previously merged model $\mathcal{M}_i$ to form a new model $\mathcal{M}_{i+1} = \{\mathcal{M}_i + \mathcal{B}_i\}$. At this point, $\mathcal{M}_{i+1}$ becomes the new baseline to detect $X_{OOD}^{i+1}$. It is a sequential structure composed of $i+1$ number of the binary classifier, i.e., one for each learned OOD, and one original neural network $\mathcal{M}_0$ that predicts $N$ classes and is placed at the end of the chain. With these notations mentioned above, we introduce our solution as follows:

**Gradient Collection:** To make this merged model $\mathcal{M}_{i+1} = \{\mathcal{M}_i + \mathcal{B}_i\}$ capable of detecting the upcoming $X_{OOD}^{i+1}$, we need to first conduct the $\mathcal{D}_{IDD}^{i+1}$ estimation on $\mathcal{X}_{IDD}^{i+1} = \{\mathcal{X}_{IDD}^i, X_{OOD}^i\}$ from both $\mathcal{M}_i$ and $\mathcal{B}_i$. Only then we can start evaluating how far the $X_{OOD}^{i+1}$ deviates from the distributions in $\mathcal{D}_{IDD}^{i+1}$. Given an input $x$, we propose to collect the gradients $\nabla_c f_{\mathcal{B}_i}(x)$ and $\nabla_c f_{\mathcal{M}_i}(x)$ separately from both $\mathcal{B}_i$ and $\mathcal{M}_i$. Then we concatenate them together $\nabla_c f_{\mathcal{B}_i + \mathcal{M}_i}(x) = \nabla_c f_{\mathcal{B}_i}(x) \| \nabla_c f_{\mathcal{M}_i}(x)$. The overall gradient dimension becomes $\nabla f_{\mathcal{M}_{i+1}} = \nabla f_{\mathcal{M}_0} \| (\nabla f_{\mathcal{B}_0} \| \dots \| \nabla f_{\mathcal{B}_i})$, where $\nabla f_{\mathcal{M}_0}$ is the gradient collected from the original neural network, and $\nabla f_{\mathcal{B}_{0\dots i}}$ are the gradients from each binary classifier in the chain.

**Class Selection:** Similar as $c_{\mathcal{M}}^{predicted}$ and $c_{\mathcal{M}}^{custom}$ to $\nabla_c f_{\mathcal{M}}(x)$, here we introduce $c_{\mathcal{B}_i}^{predicted}$ and $c_{\mathcal{B}_i}^{custom}$ to control the gradient $\nabla_c f_{\mathcal{B}_i}(x)$. Since there are only two possible predictions: IDD (label 0) and OOD (label 1), we can simply make the original prediction as $c_{\mathcal{B}_i}^{predicted}$ and use the flipped result as $c_{\mathcal{B}_i}^{custom}$

$$c_{\mathcal{B}_i}^{predicted} = f_{\mathcal{B}_i}(X;\Theta) \quad ; \quad c_{\mathcal{B}_i}^{custom} = 1 - f_{\mathcal{B}_i}(X;\Theta) \tag{4}$$

Ideally we prefer to use the label $c_{\mathcal{B}_i}^{predicted}$ for all the samples $X \in \{\mathcal{X}_{IDD}^i, X_{OOD}^i\}$, so that the gradient $\nabla_c f_{\mathcal{B}_i}(x)$ will be minimized. For $X_{OOD}^{i+1}$, using $c_{\mathcal{B}_i}^{custom}$ will make their gradients stand out even more and thus, easier to be detected.

---

**Algorithm 3** Gradient-based Evaluation of IDD Distribution

1: **function** $IDD\_estimator$ $(\mathcal{X}_{IDD}^{mem}, \mathcal{M}_i)$
2:     $\mathcal{D}_{IDD} \leftarrow [\ ]$
3:     **for each** $X_{IDD}^{mem}$ $in$ $\mathcal{X}_{IDD}^{mem}$ **do**
4:         $\nabla_c f_{\mathcal{M}_i}(x_k) = \nabla_c f_{\mathcal{M}_0}(x_k) \| \dots \| \nabla_c f_{\mathcal{B}_i}(x_k)$
5:         $\hat{\mu}_c \leftarrow \frac{1}{N_c} \sum\limits_{y_k=c} \nabla_c f_{\mathcal{M}_i}(x_k)$
6:         $\hat{\Sigma}_c \leftarrow \frac{1}{N_c} \sum\limits_{y_k=c} (\nabla_c f_{\mathcal{M}_i}(x_k) - \hat{\mu}_c)(\nabla_c f_{\mathcal{M}_i}(x_k) - \hat{\mu}_c)^\top$ where $(x_k, y_k) \in X_{IDD}^{mem}$
7:         $\mathcal{D}_{IDD}.append((\hat{\mu}_c, \hat{\Sigma}_c))$
8:     **end for**
9:     **return** $\mathcal{D}_{IDD}$
10: **end function**

**Memory for $\mathcal{D}_{IDD}$ Evaluation:** The extra dimension from $\nabla f_{\mathcal{B}_i}$ requires the pre-estimated distributions in $\mathcal{D}_{IDD}$ to be re-evaluated. For each class in $\mathcal{X}_{IDD}^{i+1}$, we assign a memory at a constant size to re-evaluate $\mathcal{D}_{IDD}$ after the expansion of the gradient dimension. In addition, the newly learned $X_{OOD}^i$ becomes a new IDD class from the perspective of $\mathcal{M}_{i+1}$ and thus, $\mathcal{D}_{IDD}$ needs to include $D_{IDD}^i$. Before we use $\mathcal{M}_{i+1}$ to detect the next OOD, $\mathcal{D}_{IDD}$ should include the distribution estimation of $N + i + 1$ classes, in the dimension of $\nabla f_{\mathcal{M}_{i+1}}$.

Furthermore, a small memory for each class helps improve the ND accuracy. Previous methods only rely on the predicted IDD and OOD from the novelty detector engine to train the binary classifier, with limited training accuracy and convergence speed when the prediction is not accurate enough. Instead, we propose to directly use the pre-stored IDD samples as the IDD training dataset. For the OOD, we still use the predicted OOD from the ND routine. Such a solution makes the training of the binary classifier more stable, by reducing mislabeled OODs. After the system converges towards current OOD, those predicted OOD samples, selected by the novelty detector with the assistance from the binary classifier, will be stored in the memory as the representation of this newly learned class, which will be used for $\mathcal{D}_{IDD}$ evaluation and future training on the new binary classifier. Except for the first $N$ classes which have the pre-labeled dataset, all the samples from the new coming OODs are self-selected by the engine itself.

### 3.4 BATCH-MODE TRAINING AND INFERENCE

The success of the sequential classification, from a sequence of binary classifiers, relies on the high inference accuracy of each binary classifier. Even with very minor accuracy degradation from each of them, the accumulated accuracy will drop exponentially. To achieve such a goal, we propose to not only consider the sample itself but the context of the sample as well. If a testing sample is within the cluster of other samples that all come from the same class, its prediction will be biased towards that class as well. This inspires the idea of the batch-mode training and inference: Based on the features, we will cluster the samples in one class into a batch first, before feeding them to the engine. During the training phase, we introduce the new loss function as follows:

$$\mathcal{L}(\mathcal{X}) = \frac{1}{N} \sum_{i=0}^{N-1} \mathcal{L}(f(X^i), i) \tag{5}$$

where each $X^i$ refers to a single batch with samples of class $i$ only. Different from traditional training method where the feed-forward operation is conducted on a single batch that contains randomly selected samples from $N$ classes, we send $N$ individual one-class batches to the classifier all at once and calculate their average loss. Due to the nature of the BatchNorm layer in the neural network, we find that the batch average from each class can be better separated and thus, the boundary of the classes is easier to be learned.

To prove the effectiveness of the batch-mode training, we evaluate the feature distribution of IDD and OOD using both the traditional and batch-mode method with CIFAR-10 dataset on a binary classifier, where IDD contains five classes and OOD contains one class. The batch size is at 32 samples. As shown in Fig. 2(a)(b), the result from batch-mode training is less intertwined, promising a higher chance to detect OOD. We exploit this property to improve the accuracy of each binary classifier.

One challenges in this approach is that only $\mathcal{X}_{IDD}$ classes are available for creating the pure batches, but the OOD samples are mixed with IDD in the incoming stream, not pre-labeled. Therefore, a pre-filtering operation is necessary to detect and prepare the OOD from the data stream. We propose to first divide the input stream into small consecutive batches and use a purity metric to localize the batches with the highest OOD percentage. In reality, the assumption of IDD/OOD batches is quite feasible. For instance, in a video or audio stream, once OOD data appear, there will be multiple, continuous samples of OOD, rather than one glitching sample only.

Regarding the purity metric, we propose to compare the mean of the features from the testing batch with the pre-estimated features of each class in $\mathcal{X}_{IDD}$. More specifically, assuming the system is in status $\mathcal{S}_i = \{\mathcal{X}_{IDD}^i, X_{OOD}^i\}$, there are totally $N + i$ classes in $\mathcal{X}_{IDD}^i$. Therefore, we expect the next input stream will contain a mixture of $X_{OOD}^i$ batch and $N + i$ kinds of batches from previous classes. We use $\mathcal{M}_0$ to filter out the batches that contain the first $N$ IDD classes, by comparing the $L_2$ distance between the batch features $\psi_{\mathcal{M}_0}(X_{test})$ and the $\mathcal{X}_0^{IDD}$ features $\psi_{\mathcal{M}_0}(\mathcal{X}_0^{IDD})$, both extracted from $\mathcal{M}_0$. Since $\mathcal{M}_0$ is trained by $\mathcal{X}_0^{IDD}$, if $X_{test}$ batch is dominated by the samples from $\mathcal{X}_0^{IDD}$, then the $L_2$ distance between them should not be large. On the other hand, if the batch is mixed with samples from another class, then the $L_2$ distance will increase which helps us set a threshold to separate them. The selected batches will be sent to the upstream of $M_0$, which is $B_0$, for another round of filtering to remove the batches that are dominated by $X_{OOD}^0$. This procedure continues along the sequential structure from bottom to top until all the batches containing the $N+i$ classes are filtered out. The remaining batches then only contain the new OOD $X_{OOD}^i$.

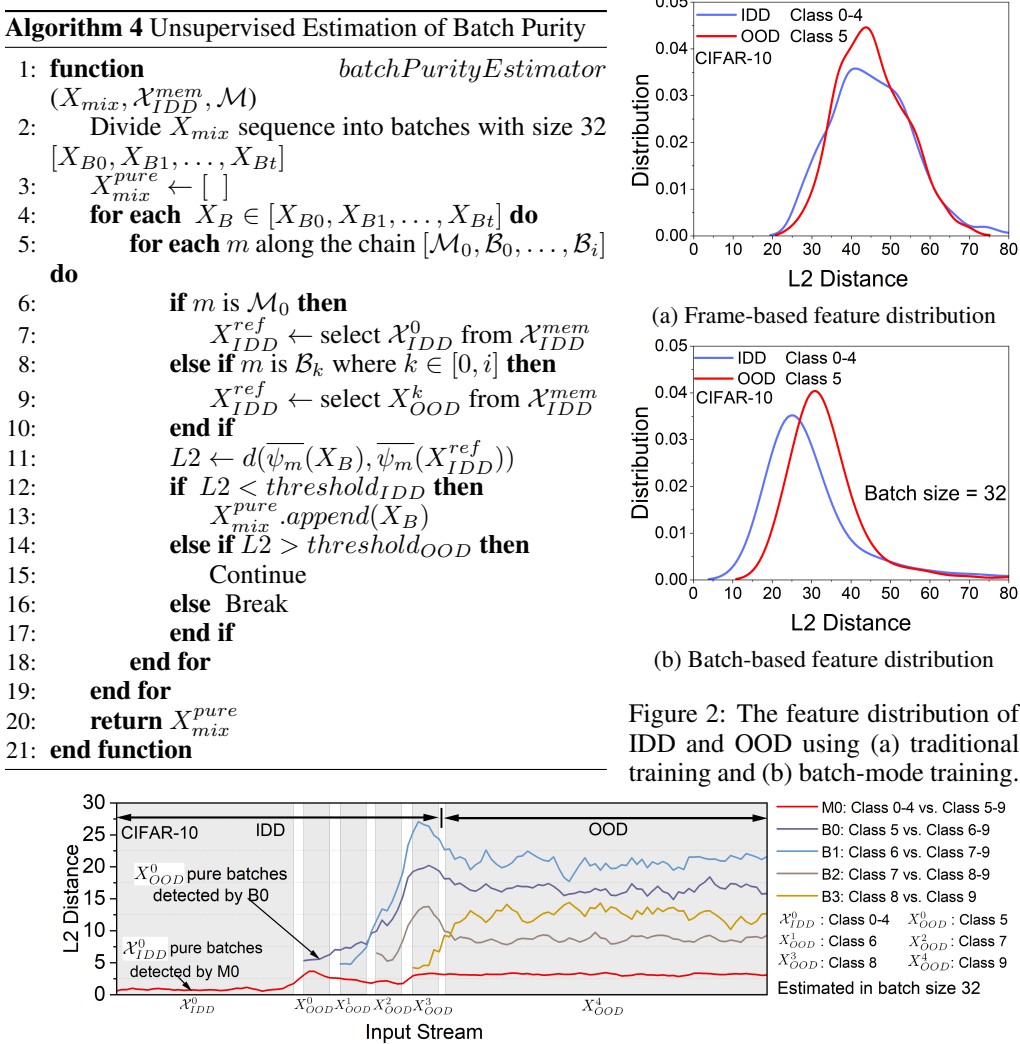

**Algorithm 4** Unsupervised Estimation of Batch Purity

1: **function** $batchPurityEstimator$ $(X_{mix}, \mathcal{X}_{IDD}^{mem}, \mathcal{M})$
2:     Divide $X_{mix}$ sequence into batches with size 32 $[X_{B0}, X_{B1}, \ldots, X_{Bt}]$
3:     $X_{mix}^{pure} \leftarrow [\ ]$
4:     **for each** $X_B \in [X_{B0}, X_{B1}, \ldots, X_{Bt}]$ **do**
5:         **for each** $m$ along the chain $[\mathcal{M}_0, \mathcal{B}_0, \ldots, \mathcal{B}_i]$ **do**
6:             **if** $m$ is $\mathcal{M}_0$ **then**
7:                 $X_{IDD}^{ref} \leftarrow$ select $\mathcal{X}_{IDD}^0$ from $\mathcal{X}_{IDD}^{mem}$
8:             **else if** $m$ is $\mathcal{B}_k$ where $k \in [0, i]$ **then**
9:                 $X_{IDD}^{ref} \leftarrow$ select $X_{OOD}^k$ from $\mathcal{X}_{IDD}^{mem}$
10:             **end if**
11:             $L2 \leftarrow d(\overline{\psi_m}(X_B), \overline{\psi_m}(X_{IDD}^{ref}))$
12:             **if** $L2 < threshold_{IDD}$ **then**
13:                 $X_{mix}^{pure}.append(X_B)$
14:             **else if** $L2 > threshold_{OOD}$ **then**
15:                 Continue
16:             **else** Break
17:             **end if**
18:         **end for**
19:     **end for**
20:     **return** $X_{mix}^{pure}$
21: **end function**

(a) Frame-based feature distribution

(b) Batch-based feature distribution

Figure 2: The feature distribution of IDD and OOD using (a) traditional training and (b) batch-mode training.

Figure 3: $L_2$ distance-based batch purity estimation using $\mathcal{M}_0$ and $\mathcal{B}_{0-3}$ on IDD: $\{\mathcal{X}_{IDD}^0, X_{OOD}^0, \ldots, X_{OOD}^3\}$ and OOD: $X_{OOD}^4$. All data are collected using CIFAR-10 dataset. Fig. 3 illustrates the process to separate the pure batches from the CIFAR-10 input stream that consists of five types of IDDs ($\{\mathcal{X}_{IDD}^0, X_{OOD}^0, \ldots, X_{OOD}^3\}$) and one OOD ($X_{OOD}^4$), using $\mathcal{M}_0$ and $\mathcal{B}_{0-3}$. The gray and white area corresponds to the batches with 100% purity and mixture data, respectively. Starting from $\mathcal{M}_0$ (the red curve), $\mathcal{X}_{IDD}^0$ pure batches are collected by comparing the $L_2$ distance with low threshold. The batches above threshold are sent to $\mathcal{B}_0$ to find the $X_{OOD}^0$ pure batches. This process continues until the stream data reaches $\mathcal{B}_3$ and all IDD and OOD pure batches are successfully separated.

## 4 EXPERIMENTS

To prove the efficacy of our proposed method, we conduct several one-class ND and CL experiments using MNIST Deng (2012), CIFAR-10, CIFAR-100 Krizhevsky & Hinton (2009) and Tiny-ImageNet Le & Yang (2015). All experiments are implemented using PyTorch Paszke et al. (2019) on NVIDIA GeForce RTX 2080 platform.

### 4.1 EXPERIMENTAL SETUP

**Input Sequence and Memory Budget:** Different from the multiple-class based setup, our method is evaluated after every exposure to a new OOD class. For every new $X_{OOD}^i$, we mix it with the same amount of randomly selected samples from $\mathcal{X}_{IDD}^i$. This mixed input stream is then sent to the system for ND and CL. The input stream consists of the batches from $N + i$ classes in $\mathcal{X}_{IDD}^i$ and

from the $X_{OOD}^i$ current class. Each batch is at the size of 32 frames for all the experiments. We also create the transition phases to mimic the input change from one class to another. Each transition phase last three batches long. The ratios of the mixture between the previous class and the next class are 1/4, 1/2 and 3/4. This transition setup is used in batch purity evaluation. For each dataset, the size of $X_{OOD}^i$ and $\mathcal{X}_{IDD}^i$ samples is shown in the first row of Table 1. $X_{OOD}^i$ samples are selected from the training dataset while the $\mathcal{X}_{IDD}^i$ are selected from the testing dataset. The reason for using testing rather than training on $\mathcal{X}_{IDD}^i$ is because previously trained binary classifiers have already seen the training data of $\mathcal{X}_{IDD}^i$ and we need to avoid any unfair evaluation in the current iteration. The second row of Table 1 presents the memory budget for $X_{IDD}^0$.

**Network Structure and Training:** For fair comparisons with previous methods, we select the structure of $\mathcal{M}_0$ as shown in Table 1, Row 3. The binary classifier has three convolution layers, one BatchNorm layer, and a Sigmoid classifier. This structure is used in all experiments. For $\mathcal{M}_0$ training, standard Stochastic Gradient Decent is used with momentum 0.9 and weight decay 0.0005. The number of epochs is listed in Table 1, Row 4. The initial learning rate is set to 0.1 and is divided by 10 after reaching the 50% and 75% milestones. For the binary classifier, we train it in 100 epochs with the Adam optimizer Kingma & Ba (2014) where the initial learning rate is set to 0.0002 and the decay rate is controlled by $\beta_1 = 0.5$ and $\beta_2 = 0.999$. To estimate the novelty in the gradient space, we collect the gradients from the last convolution layer of $\mathcal{M}_0$ and the second from the last convolution layer of each binary classifier.

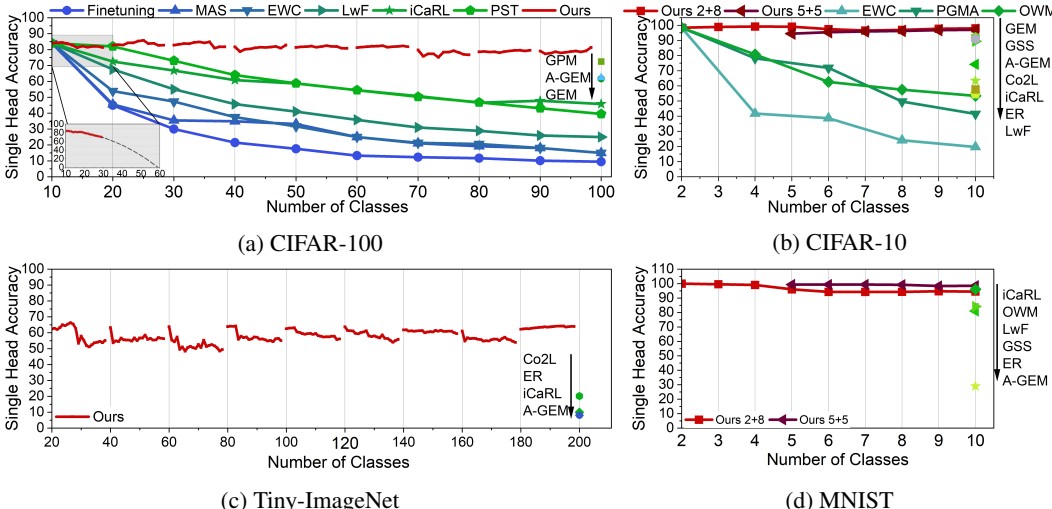

Figure 4: Single-head accuracy of one-class novelty detection and continual learning using (a) CIFAR-100, (b) Tiny-ImageNet, (c) CIFAR-10, and (d) MNIST.

## 4.2 IN-DEPTH ANALYSIS

All experiments are conducted in an unsupervised manner, which means all new OOD classes are not manually labeled, but purely rely on the prediction from the novelty detector engine. Compared with traditional CL algorithms which learn multiple classes in one shot, our method learns one class at each time and takes more steps to achieve the same learning goal. To test how many classes can be learned before the accuracy starts to drop, we test our algorithm using CIFAR-100 by training a baseline model using 10 classes and then continually feeding 20 new classes to the system one class after another. The inset from the Figure 4(a) illustrates the accuracy curve from the actual testing result (red points) plus the extrapolation (dashed line). This curve proves that our method is able to stably learn multiple steps; the accuracy eventually drops as the error in each binary classifier accumulates through the sequential process.

Therefore, we design our further experiment as follows: For CIFAR-100 and Tiny-ImageNet, we divide them into 10 tasks where each task contains 10 and 20 classes, respectively. After the baseline training with the first task, we test the algorithm performance by feeding the next task as the incoming OOD. Once all the classes from the new task are learned one by one, we terminate the current iteration and retrain a new baseline model using all the previous tasks. This new baseline

will then be used for learning the next available task. This process continues till all the tasks have been tested. For each experiment on CIFAR-10 and MNIST, we train two baseline models using the first two classes and the first five classes to mimic the 5-tasks (5T) and 2-tasks (2T) setup that used by other methods. We then feed the remaining classes to the baseline model to test the performance.

As shown in Fig. 4, the single-head accuracy of all the experiments stay at a high value after consecutively detecting and learning new classes from each checkpoint, which proves that our batch-mode method successfully boosts the performance of the binary classifier. For CIFAR-10 and MNIST, our method significantly improves the state-of-the-art even though it is unsupervised. For Tiny-ImageNet, the accuracy is less stable due to the increased complexity of the dataset, but overall the performance is still robust after learning 20 new classes.

# 5   ABLATION STUDY

We conduct four ablation studies to test how the performance is influenced by various input stream patterns using CIFAR-10 and CIFAR-100 dataset. First, we conduct two experiments by feeding the input with 75%/25% and 90%/10% of IDD/OOD mixture. As shown in Figs. 5(a)(b), the single-head accuracy of both experiments are worse than previous experiments using 50%/50% of IDD/OOD mixture. This is because fewer OOD samples increase the difficulty in OOD separation, especially the unsupervised training of the binary classifier. Second, we test our model with two smaller batch size for batch purity estimation. Figs. 5(c)(d) present the performance after dividing the input stream into smaller batches at the size of 16 and 8. The smaller the batch size is, the worse performance our model achieves. With fewer samples in each batch, the average feature estimated from the trained model becomes more diverse, which exacerbates the error in the purity estimation engine.

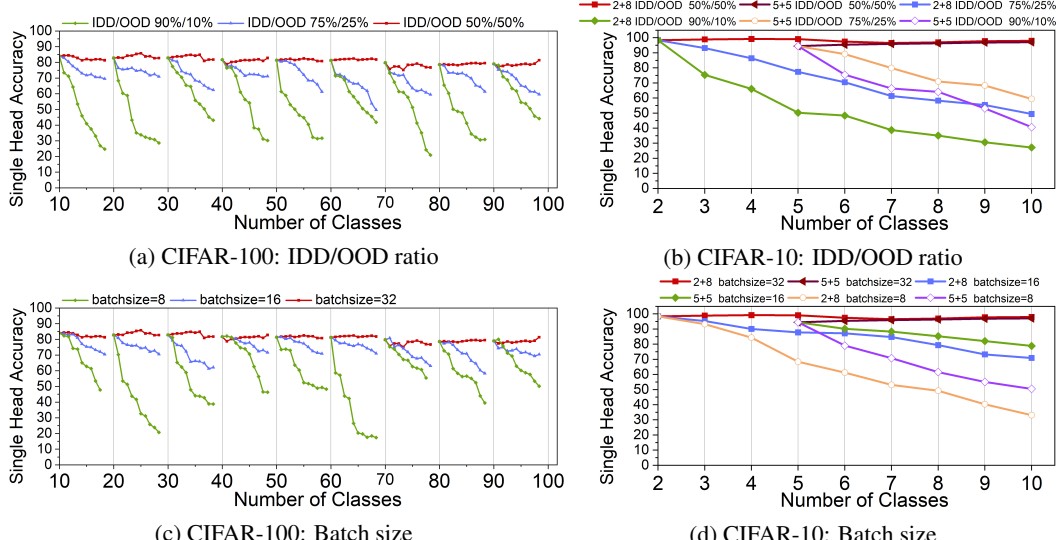

Figure 5: Ablation studies on CIFAR-10 and CIFAR-100 with (a)(b) multiple ratios of IDD/OOD mixture, and (c)(d) various batch sizes.

# 6   CONCLUSION

In this paper, we propose an unified framework for one-class novelty detection and continual learning, by using a sequence of binary classifiers with the batch-mode technique. We demonstrate that our method successfully detects and learns consecutive OOD classes in an unsupervised setup, achieving a stable single-head accuracy without triggering catastrophic forgetting. For instance, our method reaches 97.08% on CIFAR-10 in continual learning, better than the state-of-the-art. The performance on all other datasets is also among the top list, without the need of manually labeled training data. The success of this approach promises high stabilization, high learning accuracy, and practical usage by various dynamic systems.

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

## A  APPENDIX

| Parameters | MNIST | CIFAR-10 | CIFAR-100 | Tiny-ImgNet |
|---|---|---|---|---|
| Size of OOD | 4000 | 2000 | 500 | 1000 |
| Memory Budget | 4000 | 2000 | 500 | 500 |
| $\mathcal{M}_0$ | MLP | [1]ResNet-34 | ResNet-34 | ResNet-18 |
| Train Epochs on $\mathcal{M}_0$ | 30 | 70 | 200 | 200 |

[1] He et al. (2016)

Table 1: Hyper-parameters for each experiment.

