# OpenReview forum: "Self-supervised Continual Learning based on Batch-mode Novelty Detection"
_ICLR.cc/2023/Conference — Submitted to ICLR 2023_

### Official Review · Reviewer_t1EJ · 2022-10-23

**Confidence:** 4
**Correctness:** 3
**Technical Novelty And Significance:** 3
**Empirical Novelty And Significance:** Not applicable
**Recommendation:** 3

**Clarity, Quality, Novelty And Reproducibility:**

The paper is quite difficult to read due to the large amount of different mechanisms and concepts and notations (e.g. OOD becoming IID for different indices i etc.). The diagram in Fig. 1 certainly helps but maybe a second diagram (similar to Fig. 3) would be useful.
Moreover, the presentation is quite overloaded due to reduced spaces and floating/double-column algorithms and figures. Probably some parts could be put in the appendices to improve the readability.

More specifically:
- The approach uses a memory of IID samples to help the OOD. The memory budget is relatively high. This clearly helps to improve the OOD accuracy but is somewhat contrary to the continual learning paradigm (although the memory seems not to be used for further training and other methods also use a memory although at a smaller scale).
- Figure 3 is not clear. Especially the legend denoting the classes and classifiers.
- What is the impact of the number of classes for the initial classifier M_0 on the performance? Why can't we just start with 1 or 2 classes? Can this still considered continual learning?
- The paper states that the method learns one class at a time. But in the experiments, for CIFAR-100 and Tiny-ImageNet, one task is composed of 10/20 classes. What is the reason? Is this done in the same way for the state-of-the-art methods?
- It is not clear what Figure 4 represents. What do you mean by "Single Head Accuracy"? What is the overall accuracy after training N classes?
- The scalability of the approach may be questionable, both in terms of memory and computing requirements. For each new task/class a small CNN for binary classification is added. The authors only state that the CNN produces little overhead compared to other existing CL methods. It is a 3-layer CNN with one FC but The exact architecture or number of parameters is not specified. For example for CIFAR-100, if one task would be one class, we would end up with (almost) 100 CNNs that would need to by executed sequentially for inference.


**Strength And Weaknesses:**

Strengths:
- Original approach for continual learning

Weaknesses:
- Paper difficult to read
- Overloaded presentation and results difficult to interpret
- Scalability may be an issue

**Summary Of The Paper:**

This paper presents a new method for continual learning with convolutional neural networks for image classification which is based on a novelty detection mechanism using gradients (Sun et al. 2022) and a iterative training and adding of binary IDD/OOD classifiers as new classes arrive in the input stream.
The accuracy of OOD is further improved by a batch-mode estimation where a pre-filtering step determine the most pure OOD batches from the mixed IDD/OOD batches.
The experimental results show that this approach is able to achieve a high performance with a large number of classes.


**Summary Of The Review:**

Overall the paper proposes a novel and interesting approach for continual learning. But the presentation is lacking in some aspects making the paper difficult to read and understand.
Also the fact that a large memory buffer and an initial classifier is used for the first N classes may make the approach less suitable for real "continual learning" and potentially not comparable to the state of the art in the domain.

---

### Official Review · Reviewer_wx2w · 2022-10-24

**Confidence:** 5
**Clarity, Quality, Novelty And Reproducibility:** The review above elaborates on all th…
**Correctness:** 2
**Technical Novelty And Significance:** 2
**Empirical Novelty And Significance:** 2
**Recommendation:** 3

**Strength And Weaknesses:**

## Originality and Quality
### Strengths
* The paper attempts to unify novelty detection and continual learning techniques, which are novel and might interest the community.
* The experiments are conducted in an unsupervised setting, a comparatively more realistic setting for practical applications.
## Weaknesses
* The proposed framework resembles the existing task-free continual learning setting [1, 2], where it uses the gradient-based novelty detector to detect the OOD/new task examples. Thus, the novelty of the framework is limited. Further, it does not include a discussion and comparison with this setting.
* While the paper focuses on the self-supervised CL setting, it does not compare or mention the prior methods [3, 4] that have investigated self-supervised continual learning.
* The paper also needs a comparison with recent supervised CL methods [5, 6], which makes it challenging to evaluate the utility of the proposed method.
* Many design choices and hyper-parameters need to be better ablated and justified. For example, the selection of the linear classifier architecture, learning rate, epochs used for training it (ensuring fair comparison here in comparison to without it), the effect of batch size in Figure 5, and its effect on performance.

---

## Clarity
Overall, the algorithm and method are well-written and easy to follow. These suggestions should improve the paper's readability and overall presentation.
* The details about the baselines should have been provided. Further, the information on the dataset split should be included when the datasets are described in the experiments section.
* The notations in the algorithm can be improved significantly. For instance, use \text when writing text in the algorithms, and use $\text{argmin}, \text{argmax}$, and $\text{softmax}$ for better presentation.
* The zoom in Figure 4a) highlights the proposed method instead of the comparison. In Figure 4c), the arrow disrupts the interpretation of other methods.
* Many references do not reflect the conference proceedings bibliography.

---

## Reproducibility
 The code is not provided with the submission. Since the paper is empirical, it is necessary to provide the code to aid the reproducibility of future works.

---
## References
[1] Aljundi et al, Task-Free Continual Learning. CVPR 2019.
[2] Wang et al. Improving Task-free Continual Learning by Distributionally Robust Memory Evolution. ICML 2022.
[3] Madaan et al. Representational Continuity for Unsupervised Continual Learning. ICLR 2022.
[4] Fini et al. Self-Supervised Models are Continual Learners. CVPR 2022.
[5] Buzzega et al. Dark Experience for General Continual Learning: a Strong, Simple Baseline. NeurIPS 2020.
[6] Saha et al. Gradient Projection Memory for Continual Learning. ICLR 2021.

**Summary Of The Paper:**

The paper focuses on unifying novelty detection and continual learning methods by extracting and learning from the OOD data and merging it with the IDD distribution. It uses a sequence of binary classifiers with a batch-mode technique to achieve this goal. The proposed method improves performance compared to prior CL methods on MNIST, CIFAR-10/100, and Tiny-ImageNet datasets.

**Summary Of The Review:**

The paper proposes an interesting combination of novelty detection and continual learning techniques; however, due to the missing discussion and comparison with prior task-free CL and self-supervised CL methods, it is difficult to position the work. Furthermore, the experimental evaluation is weak and lacks comparison with recent CL methods. Therefore, for these reasons, the paper is not ready for acceptance in its current form.

---

### Official Review · Reviewer_PhMt · 2022-10-27

**Confidence:** 4
**Correctness:** 3
**Technical Novelty And Significance:** 2
**Empirical Novelty And Significance:** 2
**Recommendation:** 3

**Clarity, Quality, Novelty And Reproducibility:**

The description of the algorithm is not crystal clear (e.g., threshold choosing rule), and the main methods are also adopted from previous publications.

**Strength And Weaknesses:**

Strength
- The motivation of the paper is rooted in the practical scenario in which OOD data is continuously arriving while learning IDD data.
- The paper attempted to merge the two method from ND and CL.
- Gradient based novelty detection is adopted for the continual learning scenario.

Weakness
- The proposed method applies several heuristics, which is not clear whether it can be really generalized to settings other than the ones considered in the paper. For example, the threshold for novelty detection (Alg. 2) and threshold_OOD (in Alg.4) should play important roles on overall performance, but it is not clear how to choose them. If the thresholds were chosen to maximize the overall test performance, it violates the continual learning assumption.
- CIFAR-10/100, MNIST seem to be too small/standard benchmarks, and testing on more enlarged / diverse datasets should be necessary, provided that one of the main concern of the paper is OOD novelty detection. The new class in CIFAR could be thought of as a novel data, but they might have still similar distribution. What happens when completely differently distributed data point comes? From this reasoning, I think the current form of the paper is somewhat limited for a publication.

**Summary Of The Paper:**

The paper proposes a novelty detection based continual learning method. The scenario is practical, but the proposed method seems to be largely heuristic and the tested scenario is somewhat contrived.

**Summary Of The Review:**

From above reasoning, I think the paper considers an important variation of ND + CL setting, but the overall scheme requires many heuristics and it is not clear how they will generalize to other data settings.

---

### Official Review · Reviewer_e2pH · 2022-10-31

**Confidence:** 4
**Correctness:** 3
**Technical Novelty And Significance:** 2
**Empirical Novelty And Significance:** 2
**Recommendation:** 5

**Clarity, Quality, Novelty And Reproducibility:**

The paper presents the implementation details, which may be helpful to reproduce the reported results in the paper.
The paper lacks empirical discussion about the new contributions proposed in this work.
The novelty is limited as most of the algorithms are built on top of existing ones.

**Strength And Weaknesses:**

The motivation behind this work is convincing and interesting. Novelty detection and continual learning are really two related tasks in an open-world scenario, because most continual learning methods are limited under a closed world setting where all the incremental classes are fully labelled in a supervised learning fashion. However, there are still several concerns on this work:

-- Although novelty detection is a binary classification task in general, one-class continual learning is not so practical in the applications. In contrast, the continual learner should be able to learn more classes during each incremental session. One potential solution is to consider the task of novel class discovery which learns to cluster the unlabeled samples from a set of classes.

-- In this work, the authors discussed that the proposed method helps achieve the high accuracy in OOD detection and prediction in a
scenario where IDDs and OODs are streaming into the system, such as videos and audios. However, there has no experiments on video and audio datasets. It lacks experiments to support the conclusion above.

-- The proposed method is built based on existing novelty detection and continual learning methods. It needs more insightful clarifications on the new technical contributions in this work. Otherwise, the novelty would be limited.

-- The training procedure might be expensive when reading the Algorithms in the paper. Apart from the accuracy performance, it is encouraged to consider the computational efficiency, especially for a continual learning model.



**Summary Of The Paper:**

This paper presents a new learner which can address novelty detection and continual learning jointly. Specifically, a batch-mode training and inference method is used to maximize the feature separation between OOD and IDD data samples. Experimental results on several continual learning benchmarks show the effectiveness of the proposed method.

**Summary Of The Review:**

It is necessary to see the replies from the authors for the questions above.

---

### Decision · Program_Chairs · 2023-01-20

**Decision:**

Reject

**Justification For Why Not Higher Score:**

Reviewer opinion was consistent and unanimous on the key points of Clarity, Novelty, and the Heuristic Nature of the proposed approach. While the paper has interesting ideas, and the proposed scenario is interesting, in its current form it does not meet the bar for acceptance at ICLR.

**Justification For Why Not Lower Score:**

N/A

**Metareview: Summary, Strengths And Weaknesses:**

# Summary of Contribution

This paper describes an approach to continual learning that uses gradient-based novelty detection. The authors propose to train a simple, binary ID/OD classifier for each new class during incremental learning. They further propose a batch-mode training procedure that selects reliable OOD batches. Experimental results are given on MNIST, CIFAR-10, CIFAR-100 and TinyImageNet.

# Strengths

The main strength of this paper is its attempt to unify novelty detection and continual learning -- arguably a more realistic scenario for continual learning than other standard benchmarking scenarios. The experimental results provide some empirical evidence for the effectiveness of the proposed approach.

# Weaknesses

+ **Clarity**: The technical development in the paper is very difficult to follow given the many different concepts, notations, and verbose super- and sub-scripts in the mathematical notation and algorithmic descriptions. The manuscript is in need of heavy revision for clarity. In its current state I feel that it would be quite difficult to reproduce the reported results on the basis of the technical descriptions in the manuscript.

+ **Novelty**: The proposed approach is largely based on existing novelty detection and continual learning techniques. Moreover, as pointed out by reviewers, the experiments are lacking comparison with recent supervised continual learning methods.

+ **Heuristics**: The proposed approach employs heuristics throughout, with the threshold used for novelty detection the most obvious one. It is unclear how to select such a threshold and whether the approach will generalize to other scenarios.

# Summary

The reviewers are nearly unanimous in their opinion that the paper is in need of revision and clarification of the novel aspects of the approach before meeting the bar for acceptance at ICLR. Moreover, all reviewers point to the heuristic nature of many of the technical continutions, and several reviewers question the computational cost and scalability of the approach to longer sequences. The author responses during the discussion period did not adequately address these critical points.